# Functional Profile of Antigen Specific CD4^+^ T Cells in the Immune Response to Phospholipase A1 Allergen from *Polybia paulista* Venom

**DOI:** 10.3390/toxins12060379

**Published:** 2020-06-08

**Authors:** Luís Gustavo Romani Fernandes, Amilcar Perez-Riverol, Murilo Luiz Bazon, Débora Moitinho Abram, Márcia Regina Brochetto-Braga, Ricardo de Lima Zollner

**Affiliations:** 1Laboratory of Translational Immunology, Medicine School, FCM, University of Campinas (UNICAMP), Campinas 13083-888, Brazil; deboramoitinhoabra@gmail.com (D.M.A.); zollner@unicamp.br (R.d.L.Z.); 2Center of the Study of Social Insects, Department of General and Applied Biology, Institute of Biosciences of Rio Claro, Institute of Biosciences of Rio Claro, São Paulo State University, Rio Claro 13500, Brazil; aperezriverol@gmail.com; 3Laboratory of Arthropods Molecular Biology - LBMA-IBRC-UNESP (São Paulo State University), Rio Claro 13506-900, Brazil; bazonmurilo@gmail.com (M.L.B.); mrb.braga@unesp.br (M.R.B.-B.); 4Venoms and Venomous Animal Studies Center-CEVAP, São Paulo State University (UNESP), Botucatu 18610-307, Brazil

**Keywords:** Hymenoptera venom allergy, *Polybia paulista*, CD4^+^ T cells

## Abstract

Insect venom can cause systemic allergic reactions, including anaphylaxis. Improvements in diagnosis and venom immunotherapy (VIT) are based on a better understanding of an immunological response triggered by venom allergens. Previously, we demonstrated that the recombinant phospholipase A1 (rPoly p 1) from *Polybia paulista* wasp venom induces specific IgE and IgG antibodies in sensitized mice, which recognized the native allergen. Here, we addressed the T cell immune response of rPoly p 1-sensitized BALB/c mice. Cultures of splenocytes were stimulated with *Polybia paulista* venom extract and the proliferation of CD8^+^ and CD4^+^ T cells and the frequency of T regulatory cells (Tregs) populations were assessed by flow cytometry. Cytokines were quantified in cell culture supernatants in ELISA assays. The in vitro stimulation of T cells from sensitized mice induces a significant proliferation of CD4^+^ T cells, but not of CD8^+^ T cells. The cytokine pattern showed a high concentration of IFN-γ and IL-6, and no significant differences to IL-4, IL-1β and TGF-β1 production. In addition, the rPoly p 1 group showed a pronounced expansion of CD4^+^CD25^+^FoxP3^+^ and CD4^+^CD25^-^FoxP3^+^ Tregs. rPoly p 1 sensitization induces a Th1/Treg profile in CD4^+^ T cell subset, suggesting its potential use in wasp venom immunotherapy.

## 1. Introduction

Insect sting allergy accounts for 14% of severe cases of allergic reactions in Latin America, resulting in considerable morbidity and significant expenses for the treatment of patients [1]. The most clinically relevant venom allergies involve the Hymenoptera order such as bees (Apoidea), wasps (Vespoidea) and ants (Formicidae) [2]. *Polybia paulista* (Hymenoptera: Polistinae), also known as the Southern American paper wasp, is one of the 320 neotropical social wasp species in Brazil, located mainly in the southeastern region. It presents a highly aggressive behavior and is related to most cases of allergic reactions, including anaphylaxis, involving wasp sting accidents in this region [3]. Similarly to other clinically relevant insects, the venom of *P. paulista* is a complex mix of low molecular weight compounds, linear polycationic peptides and allergens [4]. While small compounds and peptides are involved in toxic and local reactions, venom allergens often cause moderate to severe systemic hypersensitivity reactions, including life-threatening anaphylaxis.

The aetiology of the allergic response is complex and influenced by several factors, including genetic susceptibility, the route of exposure and allergen doses, and, in some cases, the structural feature of the molecule itself [5]. In this sense, the molecular and immunological characterization of the majority allergens from Hymenoptera’s venom is intensive [6]. These investigations provided the basis for the production of recombinant allergens, which could improve the Hymenoptera’s venom allergy diagnosis, such as the *Component-Resolved Diagnosis* (CRD), and reduction in the risk of anaphylactic reactions in allergen-based immunotherapy (AIT) protocols [7].

In recent years, the use of traditional and advanced proteomic approaches, bioinformatic tools and molecular biology techniques enabled the identification and isolation of the three major allergens from *P. paulista* venom: phospholipase A1 (Poly p 1) [8], hyaluronidase (Poly p 2) [9,10], and antigen 5 (Poly p 5) [11]. The PLA1 and antigen 5 are abundant, whereas hyaluronidase represents a minor venom compound. Particularly, Poly p 1 is a predominant allergen (6%–10% of venom dry weight) that shows a high sensitization prevalence in Brazilian allergic patients [12]. Insect PLA1 represents a marker allergen commonly used in component-resolved diagnosis (CRD) for the unequivocal differentiation of honeybee venom (HBV) from Vespinae or Polistinae sensitizations [13]. In addition, venom PLA1 has diagnostic value for the detection of allergic patients with negative IgE to antigen 5 [14].

Previously, we demonstrated that the expression of Poly p 1 in *E. coli* resulted in an immunologically active allergen. rPoly p 1 reactivity with sera from *P. paulista*-allergic patients was similarly compared with *P. paulista* venom extract and purified native Poly p 1 (nPoly p 1) [12]. Remarkably, we also showed that rPoly p 1 induces the activation of humoral response in BALB/c mice after intradermal immunization [15]. Sera from rPoly p 1-sensitized mice reacted with the native allergen as well as its homologous in venom from other clinically relevant Neotropical and European wasp. In contrast, the cellular response to the recombinant allergen remains largely unexplored. The study of the T-cell response and cytokines production triggered by rPoly p 1 could help understand the molecular mechanism involved in tolerance induction during VIT and the rational design of allergen-based treatment of allergic patients [16].

Venom immunotherapy (VIT) is the only intervention that provides long-term benefits in the management of allergic reactions caused by Hymenoptera stings [14,17]. For wasp venom allergy, subcutaneous immunotherapy (SIT) prevents the occurrence of severe reactions in 90%–95% of patients after a subsequent field insult [18]. Although SIT using venom extracts has proven to be a highly effective procedure to treat Hymenoptera venom allergy, meta-analysis studies have shown a significant risk of systemic adverse reactions outcomes [19]. The heterologous expression of allergens could be an alternative to overcome this problem and may lead to the development of personalized panels of allergens with a reduced incidence of severe adverse reactions and nonspecific sensitization during VIT [14]. The characterization of an immune response profile drive specific to these recombinant allergens is an important prerequisite for the development of novel systems for component-based treatment.

The generation of allergen-specific regulatory cells populations of T (Tregs) and B cells (Bregs), which mediate the suppression of allergen-specific T effector cells’ subsets, is critical for the success of VIT [20]. In the context of the complex system of Tregs, the CD4^+^ T cells expressing the transcription factor forkhead box P3 (FoxP3) are key to this effect [21,22]. The suppressive activity of Tregs in the inflammatory events of the allergic response is related to different mechanisms, such as the production of inhibitory cytokines, induction of cytolysis and metabolic disruption of effector cells, and suppression of the activation of dendritic cells (DCs) [21]. However, the immunomodulatory mechanisms elicited by VIT that are responsible for the development of tolerogenic responses to allergens remain partially unclear. 

This study investigated the capability of rPoly p 1 to induce the activation of antigen-specific T cell populations in sensitized mice. Furthermore, we evaluated the functional profile of CD4^+^ T cell population after antigenic stimulation in vitro. rPoly p 1 was able to induce a Th1/Treg profile on the CD4^+^ T cell from sensitized mice, suggesting that rPoly p 1 could be a potential candidate for the development of new strategies of novel allergen-based immunotherapy for the treatment of *P. paulista*-sensitized patients. 

## 2. Results

The immunogenicity of rPoly p 1 was assessed after intradermal injection in BALB/c mice (*n* = 6) and in vitro stimulus of spleen cells with *P. paulista* venom extract. We investigated the proliferation of the T cell subsets (CD4 and CD8) and the CD4^+^ T cell functional profile. 

### 2.1. Antigen-Specific T Cell Proliferation 

Flow cytometry analysis, using a CFSE probe and labelling with antibodies to specific surface markers, were performed to assess the antigen-specific proliferation of CD4 and CD8 T cell subsets. As demonstrated in Figure 1, we observed a significant expansion of antigen-specific CD4^+^ T cells (CSFE^low^) (panels a,b) in the rPoly p 1-sensitized mice in comparison with mice immunized with *E. coli* protein extract (control group) (17.83 ± 3.04% vs. 4.91 ± 0.12%, *p* = 0.0293). In contrast, no significant proliferation of antigen-specific CD8 T cells (panels c,d) was detected in comparison with the control (7.80 ± 1.16% vs. 5.76 ± 0.93%, *p* = 0.3153). These results indicate that rPoly p 1 preferentially induces the expansion of antigen-specific CD4^+^ T cells. 

### 2.2. Antigen-Specific CD4^+^ T Cells Functional Profile

Aiming to investigate the functional profile of the antigen-specific CD4^+^ T cells that proliferated in response to antigenic stimulation, we conducted flow cytometry assays to evaluate the frequency of the Tregs populations CD25^+^FoxP3^+^ and CD25^-^FoxP3^+^ (Figure 2, panels b and c, respectively) and the quantification of IL-6, IFN-γ, IL-1β, IL-4 and TGF-β1 cytokines in cell culture supernatants (Table 1). As demonstrated in Figure 2, rPoly p 1-sensitized mice presented increased frequency of both Tregs subsets in comparison with control: CD25^+^FoxP3^+^ (15.99 ± 2.21% vs. 6.88 ± 1.91%; *p* = 0.0076—Panel b) and CD25^-^FoxP3^+^ (8.16 ± 1.86% vs. 2.03 ± 0.70%; *p* = 0.0225—Panel c). 

The analysis of cytokine production of in vitro stimulated cells is presented in Table 1, which shows the significant production of IFN-γ and IL-6 in cultures from cells obtained from sensitized mice in comparison with the control group. However, there were no significant differences among IL-4, IL-1 and TGF-β1 productions. 

## 3. Discussion

Venom immunotherapy is the gold standard for the long-term disease-modifying treatment of HVA allergy. *Vespid* VIT leads to complete protection to anaphylaxis in about 96% of allergic patients [23,24]. However, the frequency of systemic adverse events resulting from VIT ranges between 8% and 20%, as demonstrated by multicenter studies [25]. In order to improve the safety of *Vespid* VIT, the characterization and the heterologous expression of the major allergenic proteins from Hymenoptera´s venom represent an important alternative to obtain immunotherapeutic products with good effectiveness and minimal undesirable side effects [14]. 

In our previous work, we verified the heterologous production and the allergenicity of the PLA1 allergen from *P. paulista* wasp venom [12]. The recombinant allergen showed similar reactivity compared with native Poly p 1 and was extensively recognized in immunoblotting by the antigen-specific IgE (sIgE) presented in the serum from wasp-sensitized patients [12]. Furthermore, we showed that sIgE and sIgG in the sera from rPoly p1-sensitized mice recognized relevant epitopes of homologous PLA1 in venom of the Neotropical wasps *P. lanio*, *P. scutellaris* and *P. ignobilis*, *Apoica pallens* and *Agelaia pallipes,* as well as the European paper wasp *P. dominula* [15]. The capacity of the sera from rPoly p 1-sensitized mice to recognize the natural form of PLA1 presented in venom from other wasp species suggests that this recombinant allergen can induce specific immune response directed to the shared epitopes presented in PLA1 from different wasp species. This property suggests its potential use in VIT for patients sensitized to other wasps, even for those sensitized to species from a different habitat.

In this study, we analyzed the cellular immune response induced by rPoly p 1 sensitization, in particular the activation and the functional profile of the antigen-specific T cell subpopulations. As shown in Figure 1, there was significant activation of antigen-specific CD4^+^ T cell in rPoly p 1-sensitized mice in comparison with control mice. In contrast, there was no significant activation of the CD8^+^ population. Similar results were demonstrated using recombinant allergens from other wasp species, such as antigen 5 from *Vespula vulgaris* (Ves v 5) [26], and hornet (*Dolichovespula maculata*) hyaluronidase (Dol m 2) [27] and antigen 5 (Dol m 5) [28]. Similarly to these studies, our findings show that rPoly p 1 retains the T cell epitopes of the native counterpart (nPoly p 1) [27]. Corroborating these findings, the in vitro stimulation of human T cells obtained from allergic patients to *V. vulgaris* with rVes v 5 leads to the significant proliferation of specific T cell clones that expressed TCRαβ and were predominantly CD4^+^CD8^-^ [29]. 

Since CD4^+^ T cells can differentiate in populations with different functional profiles, we analyzed the cytokines produced after in vitro stimulation of the cells with *P. paulista* venom extract. We observed that, after 96 hours of stimulation, there was an elevated production of IFN-γ but no significant production of IL-4, compared with non-sensitized mice, indicating a Th1-dominant profile in rPoly p 1 after in vitro stimulation with the antigen. Interestingly, and as mentioned before, we previously demonstrated that rPoly p 1-sensitized mice presented a significant production of sIgE and sIgG that were able to recognize Poly p 1 presented in different wasp species [15]. Corroborating these findings, Bohle et al. [29] demonstrated that the response of the antigen-specific CD4^+^ T cell clones of allergic patients stimulated in vitro with rVes v 5 were not a Th2 phenotype, evidenced by the non-significant production of IL-4. Furthermore, in some patients we detected a significant production of IFN-γ, although all patients of this study had a Ves v 5-specific IgE in their sera, indicating that T cell response to Hymenoptera allergens differs from the typical Th2-dominant profile observed for aeroallergens [29].

The activation of regulatory cells evolved in the suppression of T effector cells is one of the most significant events to evaluate in VIT immunotherapy [20,21]. Here, we demonstrated that rPoly p 1 sensitization induces significant increase in Tregs, represented by CD25^+^Foxp3^+^ and CD25^-^Foxp3^+^ subpopulations. Kerstan et al. [30] demonstrated that allergic patients submitted to subcutaneous VIT protocols using aluminum hydroxide-adsorbed wasp venom presented a significant activation of CD4^+^CD25^+^Foxp3^+^ Tregs, which, by the increase in lymph-node-homing receptors such as CD62L, CD11a and CCR7, preferentially migrates to secondary lymphoid organs to suppress T-cell effector responses. This fact could evidence the importance of this Treg population to the efficiency of VIT protocols. 

Moreover, we also observed a significant increase in Foxp3^+^ Tregs that do not express the CD25 marker in the CD4^+^ T cell population from rPoly p 1-sensitized mice. In this context, Coleman et al. [31] demonstrated in an experimental mouse model of *Bordetella pertussis* infection that the CD4^+^Foxp3^+^CD25^-^ Treg population could have an important immunoregulatory function suppressing CD4^+^ T effector cells, which was partially mediated by IL-10 production and not dependent on cell contact. In agreement with this, Nishioka T. et al. demonstrated that this same population could suppress T effectors cell proliferation in vitro even in aged mice [32]. 

Here, the analysis of cell culture showed a significant production of IL-6 in sensitized mice, but no differences in TGF-β1 and IL-1β, when compared with the production of splenocyte cultures in control mice. This fact could be related to the expansion of the Foxp3^+^ Tregs populations in cultures from rPoly p1-sensitized mice, since it was demonstrated in a transgenic mice model, which constitutively overexpress IL-6, that these animals presented increased CD4^+^ Foxp3^+^ cells in the thymus and spleen with a preserved suppressive function. Furthermore, the authors demonstrated that TGF-β1 induced Tregs, which was differentiated from naïve CD4^+^ T in the periphery, (iTregs), was significantly reduced in transgenic mice, whereas the natural Tregs (nTregs) were not affected, suggesting that IL-6 acts differently on iTregs and nTregs populations [33]. However, considering the limitations of the in vitro analyses presented here, and due to the high variability of phenotypes and functional mechanisms by which Tregs exert their suppressive properties, future investigations should be conducted to better evaluate the immunomodulatory effects of rPoly p 1 to induce Tregs activation.

## 4. Conclusions

We demonstrated that the recombinant form of PLA1 from *P. paulista* wasp venom was able to induce a significant expansion of antigen-specific CD4^+^ T cells, but not of CD8^+^ T cells in sensitized mice. These cells were polarized to a Th1/Treg functional profile, demonstrated by a high production of IFN-γ and by the pronounced expansion of CD4^+^CD25^+^FoxP3^+^ and CD4^+^CD25^-^FoxP3^+^ Tregs populations, respectively. These results suggest that rPoly p 1 could be a promising eligible candidate to help develop new strategies of wasp VIT, which could supplant the risk of severe adverse allergic reactions with the use of wasp crude venom extracts. 

## 5. Materials and Methods 

### 5.1. Animals

Specific Pathogen Free (SPF) female BALB/c mice (four weeks old) were obtained from the Multidisciplinary Center for Biological Research (CEMIB) of the University of Campinas (UNICAMP) Campinas, SP, Brazil. They were maintained in an SPF environment at 24 °C ± 1 for a photoperiod of 12/12 hours, and fed with autoclaved Nuvilab CR-diet and water ad libitum for at least four weeks before being used in experiments. The methods described in this manuscript were carried out in accordance with the ‘Guide for the Care and Use of Laboratory Animals,’ as promoted by the Brazilian College of Animal Experimentation (COBEA), and approved by the Ethics Committee for Animal Experimentation of the University of Campinas (CEUA/UNICAMP. Protocol #3476-1, approval date 3 September 2014). 

### 5.2. Polybia Paulista Venom Extracts and Recombinant PLA1 

The venom extracts and rPoly p 1 were obtained expressing the PLA1 cDNA cloned in specific vectors and expressed in *E. coli*-competent cells using the previously described protocols [12,15]. For the rPoly p1 purification, the soluble fractions were applied to a prepacked column HisTrap HP™ (Ni^+2^ Sepharose™ High Performance; GE Healthcare, Sweden) as described before [12], followed by SDS-PAGE (15%) analysis for monitoring the efficiency of the purification process. 

### 5.3. Mice Sensitization 

Mice sensitization to rPoly p 1 was conducted as previously described [15]. Briefly, the mice (*n* = 6) were immunized intradermally with six weekly doses of 20 µg of rPoly p 1 in phosphate-buffered saline (PBS) with 5% of Al(OH)_3_ (rPoly p 1 group). The control group (*n* = 4) was immunized with 20 µg of solubilized proteins from *E. coli* BL21 (DE3) cytoplasmatic content transformed with an empty pET-28a vector serving as negative control. Seven days after the last immunization dose, the mice from both the rPoly p1 and control group were bled and sacrificed. The sera were separated and stored at −20 °C until their use in ELISA assays, and spleens were aseptically removed for antigen-specific T cell proliferation assays. Figure 3 represents a summary of the experimental procedures. 

### 5.4. Antigen-Specific T Cell Proliferation Assay

In order to stimulate the activation of the antigen-specific T cell, proliferation assays were conducted as described below. Briefly, mice spleens were collected aseptically, and the splenocytes were obtained after homogenizing the organs in cell dissociation sieves in RPMI 1640 medium (Vitrocell, Campinas, SP, Brazil) supplemented with sodium bicarbonate (2 g/L, Sigma-Aldrich®, St. Louis, MO, USA), HEPES buffer (25mM, Sigma-Aldrich®, St. Louis, MO, USA), 10 UI/mL of penicillin, 10 μg/mL of streptomycin, and 5 μg/mL of Fungizone (all from Gibco, Invitrogen Corporation, Grand Island, NY, USA). Erythrocytes were lysed with the lysis solution (150mM NH_4_Cl, 10mM KHCO_3_ and 0.1mM EDTA) at room temperature, cells were pelleted at 400× *g* for 5 min, and cells were resuspended in RPMI cell culture medium. Splenocytes were previously stained with 2.5 µM of the probe 5(6)-CFDA; 5(6)-Carboxyfluorescein diacetate (CFSE, Biolegend, San Diego, CA, USA), seeded in 96-well plates at 2.5 × 10^5^ cells/well in the presence of 4 µg/mL of venom extracts from *P. paulista*, and then incubated for 96 h at 37 °C in humidified incubator with CO_2_ 5%. After this period, the cell cultures’ supernatants were collected for the quantification of cytokines, as described in Section 5.5, and the cells were then stained with phenotypic specific markers for CD4^+^ T cells, CD8^+^ T cells or T regulatory cells, as described below. The proliferation and the frequency of Tregs were assessed by flow cytometry, as described in Section 5.6.

### 5.5. Measurement of Cytokines

The cultures’ supernatants were collected for the dosage of cytokines through ELISA using commercial kits (ELISA Ready Set-Go™ e-Bioscience, Thermo Fischer Scientific, Waltham, MA, USA) according to the manufacturer’s recommendations. We analyzed the following mouse cytokines: IL-6, IFN-γ, IL-1β, IL-4 and TGF-β1.

### 5.6. Flow Cytometry

Following cultivation of the splenocytes in the antigen-specific T cell proliferation assays, single-cell suspensions were firstly stained with the viability probe Zoombie NIR™ fixable viability kit (BioLegend, San Diego, CA, USA). After washing with flow cytometry cell staining buffer (FCSB—phosphate-buffered saline pH 7.4 containing 5% SFB, 2mM EDTA and 2 mM NaN_3_) the cells were labelled with anti-CD4-APC (Biolegend), and anti-CD8-PercPCy5. (Biolegend). For the Tregs analysis, the cells were labelled with anti-CD4-Fitc (Biolegend) and anti-CD25-APC (Biolegend); additionally, the samples were incubated with fixation/permeabilization buffers (Foxp3 Fix/Perm Buffer Set—eBioscience-Affymetrix, Santa Clara, CA, USA), according to the manufacturer’s instructions, and stained with anti-mouse—Foxp3 PE, antibody (Biolegend). The cells were also stained with irrelevant isotype controls for each cell marker analyzed. The samples were acquired using FACsVerse™ Flow Cytometer (BD-Bioscience San Jose, CA, USA), where 10,000 events gated on viable cells for each tube were established as the minimum number of events for the acquisitions. Adjustments to the photomultiplier tubes (PMTs) voltage and compensation were determined by the examination of non-marked cells and single-stained cells using the antibodies described above. The compensation matrix was stablished manually using the FACSuite acquisition software version 1.0.6 (Becton Dickinson, San Jose, CA, USA). The results were analyzed with FCS Express 6 Flow Research Edition Software (DeNovo Software, Los Angeles, CA, USA) and the gating strategy used in the analysis of T cell proliferation and Tregs frequency are shown in Figure A1 and Figure A2 in Appendix A, respectively.

### 5.7. Statistical Analyses

Data were statistically analyzed using Mann–Whitney U test. The software Prism (version 6.0, Graphpad Software, Inc., San Diego, CA, USA) was used for statistical and graphic analysis. Significant differences were set at *p* ≤ 0.05. Data from flow cytometry analyses are represented as Tukey box plots, and the mean ± S.E.M. expresses data from cytokines quantification.

## Figures and Tables

**Figure 1 toxins-12-00379-f001:**
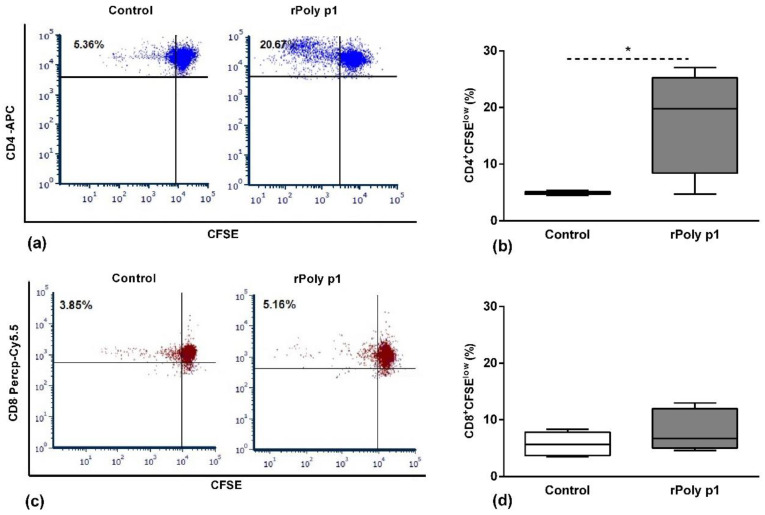
Antigen-specific T cell proliferation. Cells obtained from mice spleens were previously stained with a CFSE probe and, after the culture period, were harvested and stained with anti-CD4 (**a**,**b**) and anti-CD8 (**c**,**d**) monoclonal antibodies conjugated to fluorochromes APC and Percp-Cy5.5, respectively. Panels (**a**) and (**c**) are a representative dot plot showing the frequency (%) of CFSE^low^ (proliferating cells) in CD4 (**a**) or CD8 (**c**) T cell subsets, and the boxplots (**b**) and (**d**) represent their corresponding data, obtained in Control and rPoly p1 groups. **p* < 0.05 Mann–Whitney U test.

**Figure 2 toxins-12-00379-f002:**
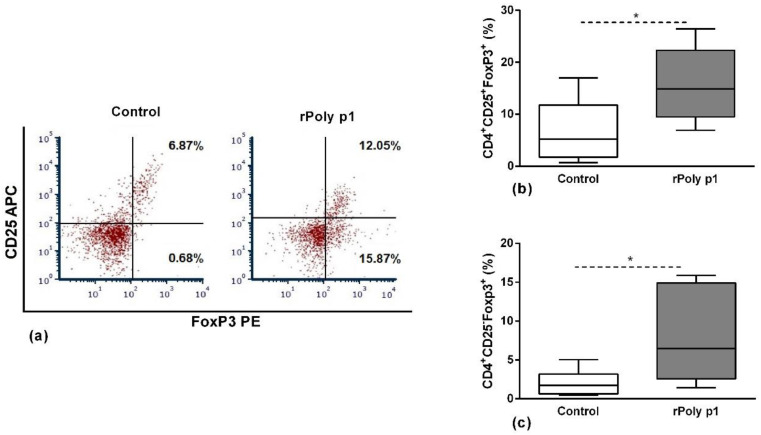
Tregs population frequency after in vitro antigenic stimulation. Cells stained with monoclonal antibodies conjugated to fluorochromes to Tregs markers: CD4, CD25 and FoxP3. Panel (**a**) presents a representative dot plot showing the frequency (%) of CD4^+^CD25^+^FoxP3^+^ Tregs (upper right quadrant) and CD4^+^CD25^-^FoxP3^+^ Tregs (lower right quadrant), and the boxplots (**b**) and (**c**) represent their corresponding data obtained in Control and rPoly p 1 groups. **p* < 0.05 Mann–Whitney U test.

**Figure 3 toxins-12-00379-f003:**
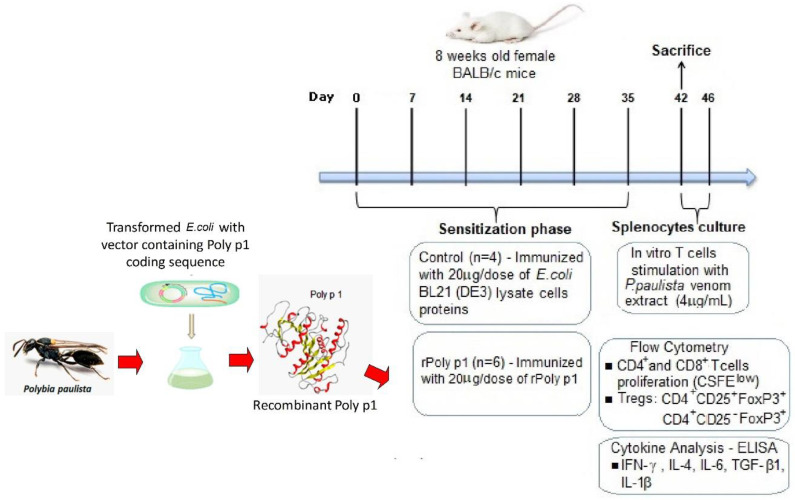
Timeline for sensitization with rPoly p 1 and experimental procedures description. Mice were sensitized to rPoly p 1 by weekly intradermal injections (six doses—20 µg/dose), the control group received a 20 µg/dose of *E. coli* BL21 (DE3) cells’ lysate proteins. Seven days after the last dose, the mice were sacrificed, and the splenocytes were cultured for 96 hours in the presence of *P. paulista* venom extract (4 µg/mL) for antigen-specific stimulation of T cells. After this period, the cells were stained for flow cytometry analysis to evaluate the proliferation of T cells and the frequency of regulatory T cells populations. The cell culture supernatants were collected for cytokines quantification by ELISA assays. *Polybia paulista* photo was kindly provided by Dr. Mario S. Palma. BALB/c mouse photo is from Staton K. Short, The Jackson Laboratory (https://www.jax.org/strain/000651).

**Table 1 toxins-12-00379-t001:** Cytokine quantification in cell culture supernatants of in vitro stimulated T cells with *Polybia paulista* venom extract.

Cytokine (pg/mL)	Control ^1^ (*n* = 4)	rPoly p1 ^1^ (*n* = 6)	Statistical Significance ^2^
IFN-γ	135.30 ± 74.67	1040 ± 172.50	*p* = 0.0006 *
IL-4	4.36 ± 0.44	4.99 ± 0.37	*p* = 0.28
IL-6	37.22 ± 19.25	265.10 ± 57.48	*p* = 0.0048 *
IL-1β	5.94 ± 0.48	6.35 ± 1.06	*p* = 0.91
TGF-β1	643.8 ± 140.4	488.3 ± 145.2	*p* = 0.474

^1^ Data presented as mean ± SEM, ^2^ Mann–Whitney U test, * Statistically significant (*p* < 0.05).

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
