# Peer review of "Functional Profile of Antigen Specific CD4+ T Cells in the Immune Response to Phospholipase A1 Allergen from Polybia paulista Venom"

_toxins, 2020, doi:10.3390/toxins12060379_

Round 1

Reviewer 1 Report

This study describes ex vivo evaluation of immune cell subsets from mice, following sensitization with wasp venom components.  In general, this brief study is well written and presented.  The findings are of importance to the field of allergy / anaphylaxis / and immunoprophylaxis.

Minor, but mandatory revisions:

  1. The authors have not shown the gating heirarchy for their flow cytometry dot plots please do so for each of the figures containing flow cytometry dot plots.
  2. How were the flow cytometry gates established?  What controls were used, such as fluorescence-minus-one for multicolor analysis etc?  Or were "isotype" controls  the only control.  Please state clearly the controls and methodologies in the methods section

Author Response

Point 1: The authors have not shown the gating hierarchy for their flow cytometry dot plots please do so for each of the figures containing flow cytometry dot plots.

Response 1: We thank the reviewer for the suggestion and agree with the suggestion to insert the flow cytometry plots demonstrating the gate strategy used. Since the article has been submitted as communication, we added this information as supplementary material in the Appendix section (figures 1A and 2A). The plots are showing the gate strategy used in the analysis of flow cytometry data from figures 1 and 2, respectively. We corrected the fluorochromes descriptions of CD4 and CD8 markers in figure 1 and CD4 and CD25 markers in figure 2, and in the manuscript text (Figure 1 legend and Material and Methods section item 5.6).

Point 2: How were the flow cytometry gates established?  What controls were used, such as fluorescence-minus-one for multicolor analysis etc?  Or were "isotype" controls the only control.  Please state clearly the controls and methodologies in the methods section

Response 2: As described on Point 1 the gate strategy of the flow cytometry analysis was provided as supplementary material in the Appendix section (figures 1A and 2A) with their respective description in figure legends. The adjusts of PMTs voltage, and the compensation was determined by the analysis of non-marked cells and single stained cells with the antibodies described on Material and Methods section. The compensation matrix was stablished manually using the FACSuite acquisition software (Becton Dickinson, San Jose, CA, USA). As recommended this information was added in the Material Material and Methods section item 5.6).   

Reviewer 2 Report

This manuscript is focused on identifying potential venom proteins which can be used for venom immunotherapy by quantifying the cytokine concentrations of recombinant phospholipase A1 from Polybia paulista induced cell lines. The manuscript was well summarized, but there are several minor points to be answered for full recommendation.

  1. Phospholipase A1 was expressed in E. coli competent cells. Are there any other reasons to use E. coli expression system than baculovirus expression system which is known to provide high quality of proteins?
  2. In Materials and Methods section (5.2., Line 223), there should be more descriptions about purifications. In venom immunotherapy, one of the most important thing for venom injection could be highly purified recombinant venom protein. Therefore, I suggest authors to put additional figures such as SDS-PAGE of Non- and purified rPol p1 and add more description of the process for purification and purification efficiency of the recombinant protein.
  3. Is there any other specific reason to choose the concentration (20 μg) of recombinant Pol p1 for immunization? It would be much better to put the explanation for usage of this concentration in Materials and Methods section (5.3., Line 226).

There are several points to be modified and corrected as below. (References section)

  1. Modify the scientific name to Italic style.

(Polybia paulista in Line 307, 311, 315, 318, 323 and 328)

(Vespula vulgaris in Line 363 and 369)

(Bordetella pertussis in Line 376)

Author Response

Point 1: Phospholipase A1 was expressed in E. coli competent cells. Are there any other reasons to use E. coli expression system than baculovirus expression system which is known to provide high quality of proteins?

Response 1: We thank the reviewer for this comment. In addition to E. coli cells, venom phospholipase A1 have been expressed in P. pastoris (BORODINA et al., 2011) and insect cells (baculovirus system) (SEISMANN et al., 2010). The major problem for the use of these eukaryotic systems is that the expression of venom PLA1 often resulted in significantly low protein yields (BORODINA et al., 2011), potentially due to the toxic effect related to phospholipase A1 enzymatic activity in cellular membranes. The production of low levels of the recombinant protein hampers the downstream analyses which include recombinant protein purification, IgE testing, and mice immunization. It is important to note that we evaluate the heterologous production of Poly p 1 in P. pastoris (manuscript under revision), but similar to the study for the PLA1 from Vespula vulgaris (Ves v 1) (BORODINA et al., 2011), the protein yields were extremely low.      

In a previous study, we already informed the heterologous expression of the phospholipase A1 (rPoly p 1) from P. paulista wasp venom and E. coli cells (PEREZ-RIVEROL et al., 2016). Remarkably, high levels (212 mg/L) of purified (95–99%) rPoly p 1, which is the same protein used in the current study, were obtained. Moreover, rPoly p 1 showed significant allergenicity as compared to the native Poly p 1 and was recognized on immunoblotting by 100% of individual sera (n=10) from Brazilian wasp-sensitized patients (PEREZ-RIVEROL et al., 2016). Moreover, in a different study, we showed that this rPoly p 1 induces a strong humoral response in sensitized mice (PEREZ-RIVEROL et al., 2018). To summarize, based our previous results related to high protein yields, the production of highly purified protein which was recognized by the IgE sera of sensitized patient and in addition, induced a strong humoral response (allergen-specific IgG and IgE) (PEREZ-RIVEROL et al., 2018), we decided to use the rPoly p 1 produced in E. coli cells to evaluate the cellular response.    

Point 2: In Materials and Methods section (5.2., Line 223), there should be more descriptions about purifications. In venom immunotherapy, one of the most important thing for venom injection could be highly purified recombinant venom protein. Therefore, I suggest the authors put additional figures such as SDS-PAGE of Non- and purified rPol p1 and add more description of the process for purification and purification efficiency of the recombinant protein.

Response 2: We thank the reviewer for the suggestion and agree with the relevance of using highly purified recombinant allergens for this study and particularly, for the venom immunotherapy. The purification of the rPoly p 1 used in the current study was conducted in a commercial prepacked column, HisTrap HP™ Ni2+ Sepharose™ High Performance (GE Healthcare, Sweden) and was described in a previous study (PEREZ-RIVEROL et al., 2016). We include additional information with more descriptions about the purification process of the recombinant protein in the revised version of the manuscript in item 5.2 of the Material and Methods section.

Regarding the purification figures, we wanted to note that in Figure 5 of that previously published contribution (PEREZ-RIVEROL et al., 2016), we showed the results of the rPoly p 1 purification which is the same batch used in the current study, as well as in the analysis of the humoral response, also published (PEREZ-RIVEROL et al., 2018). We used the same batch in an attend to homogenize as much as possible the experimental setting of the studies.

As the purification was already described and the results are published, we respectfully suggest as an alternative to clear refer the purification data as follow:

Figure 5, Toxicon. Perez, Riverol et al., 2016

Point 3: Is there any other specific reason to choose the concentration (20 μg) of recombinant Pol p1 for immunization? It would be much better to put the explanation for usage of this concentration in Materials and Methods section (5.3., Line 226).

There are several points to be modified and corrected as below. (References section)

Modify the scientific name to Italic style.

(Polybia paulista in Line 307, 311, 315, 318, 323 and 328)

(Vespula vulgaris in Line 363 and 369)

(Bordetella pertussis in Line 376)

Response 1: We determined the immunization protocol empirically, since literature data present different protocols using different doses range and immunization routes (Winkler et al., 2003; Kouchomian et al. 1995). As we demonstrated in a previous work Perez-Riverol et. Al., 2018) this protocol was able to induce a strong antigen-specific antibody response shown sIgE and sIgG titers in ELISA assays (Figure 5). 

The scientific names presented in References section were modified to Italic style as recommended, thanks for the careful observation.

References

BORODINA, I. et al. Expression of Enzymatically Inactive Wasp Venom Phospholipase A1 in Pichia pastoris. PLoS ONE, v. 6, n. 6, 2011.

PEREZ-RIVEROL, A. et al. Molecular cloning, expression and IgE-immunoreactivity of phospholipase A1, a major allergen from Polybia paulista (Hymenoptera: Vespidae) venom. Toxicon, v. 124, p. 44–52, 2016.

PEREZ-RIVEROL, A. et al. Phospholipase A1-based cross-reactivity among venoms of clinically relevant Hymenoptera from Neotropical and temperate regions. Molecular Immunology, v. 93, n. September 2017, p. 87–93, 2018.

SEISMANN, H. et al. Recombinant phospholipase A1 (Ves v 1) from yellow jacket venom for improved diagnosis of hymenoptera venom hypersensitivity. Clinical and Molecular Allergy, v. 8, p. 7, 2010.

Winkler, B.; et. al. Allergen-specific immunosuppression by mucosal treatment with recombinant Ves v 5, a major allergen of Vespula vulgaris venom, in a murine model of wasp venom allergy. Immunology 2003.

Kochoumian, L.; Lu, G. Murine T and B cell responses to natural and recombinant hornet venom allergen Dol m 5.02 and its recombinant fragments. J. Immunol. 1995